# Nudging policymakers on gendered impacts of policy

Lindsay Blair Bochon[1], Janet Dean[1], Tanja Rosteck[1], Jiaying Zhao[2,3]*

1 Sauder School of Business, University of British Columbia, Vancouver, British Columbia, Canada,
2 Department of Psychology, University of British Columbia, Vancouver, British Columbia, Canada,
3 Institute for Resources, Environment and Sustainability, University of British Columbia, Vancouver, British Columbia, Canada

* jiayingz@psych.ubc.ca

**Data Availability Statement:** All data files are available on the Open Science Framework (OSF) database (https://osf.io/ht3yn).

**Funding:** The author(s) received no specific funding for this work.

## Abstract

Despite the proliferation of nudge research in the last few decades, very little published work aims to nudge the behavior of policymakers. Here we explore the impact of a well-established nudge on policymakers in the Northwest Territories of Canada. In a pre-registered randomized controlled trial, we emailed an invitation to policymakers ($N = 263$) to attend an online briefing on gendered impacts of policy. In the treatment condition ($N = 133$), the invitation contained personal stories of two women whose lives were disproportionally impacted by public policies more than men. In the control condition ($N = 130$), the invitation did not contain such stories. After the briefing, we sent all participants in both conditions a link to a public pledge that they could sign. The pledge was to lead and advocate for equity-oriented policymaking. Contrary to our prediction, there was a small backfiring effect where policymakers in the treatment condition (3.0%) were less likely to attend the briefing than the control condition (7.7%). However, two policymakers (1.5%) in the treatment condition signed the public pledge compared to one (0.8%) in the control condition. The current findings reveal the limits of using personal stories as a nudge to influence policymakers. We discuss insights gained from this experiment and follow-up debriefings with policymakers on how to improve future behavioral interventions designed to nudge policymakers.

## Introduction

To date, most research in the nudge literature has focused on changing the behaviors of citizens; much less work in comparison has examined the impact of nudge on policymakers' decisions [1, 2]. Elected politicians have been shown to be equally or more susceptible than citizens to the sunk-cost fallacy, the status-quo bias, temporal discounting, and risk-seeking during uncertain policy decisions [3]. The degree to which policymakers digress from rational decision making may have important implications for policymaking. This raises the question about whether it is time to reorient the focus of nudge onto government itself—nudging policymakers directly to improve the way that policy is made [1].

**Competing interests:** The authors have declared that no competing interests exist.

Gender equity in policy is one area where nudge may help. In 2015, the Canadian federal government adopted the United Nations Commission on the Status of Women's Beijing Declaration and Platform for Action. As part of this declaration, Canada made a commitment to "gender mainstreaming" ensuring that attention to gender equity is central to all governmental activities [4]. Canadian territorial governments, however, never built this goal into their legislative systems, meaning that there is no overarching requirement to conduct gender-based analysis on new and existing policies. As a result, issues of gender are rarely considered in policymaking, and policies often have unequal impacts on different genders.

Gender equity may have been overlooked by policymakers due to high information processing demands associated with the policymaking environment, where information needs to be gathered and processed quickly. Policymakers simply do not have the time and attentional capacity to learn about and consider every policy issue [5]. Other persistent barriers include the lack of enforceable requirements towards gender mainstreaming, often resulting in 'lip service' towards the principles of gender equality without the implementation of any practical changes [6]. Gender mainstreaming also challenges the status quo, resulting in limited implementation of gender equity strategies that challenge existing power structures [6, 7]. Despite these barriers, however, Payne [6] suggests that professional networks and authoritative experts can engage with policymakers in a number of formal and informal ways to influence policy development. For example, presenting policymakers with information in a short, accessible format is often desirable, which reduces information complexity and facilitates decision making [5, 8, 9]. One effective way to do this is to provide policymakers with a policy briefing, which is a non-technical synthesis of an issue intended to influence decision making about complex policies [10]. Policy briefings are a common method of communication with policymakers, and have been shown to be effective at increasing awareness and influencing policymakers' beliefs about target issues [10, 11].

Additionally, since politicians tend to be reliant on public opinion for re-election, increasing the salience of political accountability may be an effective way of increasing commitment toward particular issues. A public pledge is a specific strategy for increasing policymakers' commitment through either verbal or written promise to act, binding a policymaker to a particular behavior and increasing their self-expectations for engaging in that behavior [12–14]. Public pledges are particularly effective because they increase political accountability and emotional investment with the issue [15]. Many studies have found that public pledges, both alone and combined with other interventions, can be effective in promoting a broad range of target behaviors, including pro-social behaviors such as recycling [16], towel reuse among hotel guests [17], reducing water consumption [18], energy saving [19], and health behaviors such as seat belt use [20], particularly when the pledges are made publicly instead of privately [21].

There is a growing need for Canadian territorial governments to employ methods such as policy briefings and public pledges to increase awareness, understanding, and commitment on the issue of gender equity in policy. As of 2021, the territorial government of the Northwest Territories (NWT) has identified gender equity as a key legislative priority. Furthermore, the Status of Women Council of the NWT is an organization committed to furthering gender equity in the NWT, supporting community outreach and public awareness initiatives on the issue, and working closely with municipal and territorial policymakers.

Of the limited research on nudging policymakers, a few studies showed that personal stories (i.e., narratives) provide a persuasive medium for the promotion of behavior change [22–24], in increased support for controversial political policies [25] and improving health-related behaviors [26]. Meta-analytic evidence provided by Braddock & Dillard [27] suggests that personal stories exert a causal influence on four primary indices of persuasion: beliefs, attitudes,

intentions, and behaviors; while other research [28–30] suggests there are multiple psychological routes leading to these persuasive effects.

The first route is immersion in a narrative that transports the reader into the story, such that they vicariously experience events as they unfold. Transportation influences real-world beliefs by suspending the tendency to counterargue about the veracity of the information presented in the story, thereby changing opinions to be in line with the story's message [31]. In the second route, identification with a protagonist leads to greater empathy and emotional engagement with the story, which, in turn, leads to the adoption of the protagonist's perspectives and beliefs [32]. Affective responses may mediate the effect of narratives on behavioral intention [33–35]. However, some studies suggest that personalized narratives that focus on episodic information about individual incidents may not always confer persuasive benefits, particularly when the goal is to mobilize collective action towards a social cause [36]. In such instances, personalization detracts from the larger structural cause of the problem.

A considerable body of research suggests that a single identifiable protagonist is more effective than a larger group to increase aid behavior (i.e., identifiable victim effect [37]). Theoretical accounts suggest that a variety of psychological mechanisms contribute to this phenomenon, including increased emotional reactivity [37–40], perceived impact of helping [39, 41], and perceived responsibility to help [37, 39, 42]. This is consistent with research suggesting that narratives may have an indirect effect on behavioral intentions by increasing personal norms, or a perceived personal obligation to act [28].

In the current study, we aim to draw attention to gender equity from policymakers in the NWT territorial government by inviting them to attend a policy briefing on gendered impacts of policy and to sign a public pledge to lead and advocate for equity-oriented policymaking. Given that issues of gender are often overlooked by NWT policymakers, we believe that utilizing personalized stories about the gendered impacts of policy would increase emotional engagement and personal responsibility towards the issue. In turn, increased engagement may translate into action in the form of briefing attendance and pledge signing. In the treatment condition, the invitation and briefing include two personal stories or narratives about two individuals who experienced unequal impacts from certain policies. In the control condition, no personal stories are included. We pre-registered one hypothesis, as well as several exploratory analyses at (https://osf.io/ht3yn). Our pre-registered hypothesis is that the treatment condition will have a higher rate of attendance at the briefing sessions than the control group. As exploratory analyses, we will also examine if the treatment group has a higher rate of accepting the email invitation or signing the pledge than the control group. We also note that to protect the privacy and confidentiality of our participants, the information on pledge signing was removed from the data uploaded to OSF.

## Pilot studies

We conducted two pilot studies to examine whether our interventions and measures would be appropriate for policymakers in the NWT. The first pilot study included 20 policymakers who were randomly assigned to one of four groups. In the control group, we emailed policymakers to invite them to sign a public pledge with a link to a website (https://www.noeconomicabuse. com/) where they could sign the pledge. The website was developed by the Status of Women Council of the NWT. In the story group, we sent the same email to policymakers, which also contained two personal stories of two women whose lives had been disproportionally impacted by policies. The reason for using two stories is to demonstrate the disproportionate harm and benefit from policies. One story depicted a woman who was disadvantaged by a housing policy and the other story showed another woman who was helped by a job training policy. The third

(checklist) group received the same email as in the control group but containing a checklist describing how to make the pledge. Finally, the fourth group received the email with both personal stories and the checklist.

On average, 35% of the policymakers across the four conditions clicked on the link to the website, but none of them signed the public pledge. This result gave us pause in focusing on pledge signing as the primary behavior to change. Policymakers may have been hesitant to sign the pledge because they did not have enough information on the topic to make a decision. With these considerations in mind, we conducted a second pilot study with several changes to the study design. First, we reduced the number of conditions from four to two (personal stories vs. control), to maximize the number of participants in each condition. Second, we shifted the focus of pledge signing to attending an online policy briefing. We reasoned that attending an online policy briefing may be less consequential than signing a public pledge and it also provides more information to policymakers.

In the second pilot study, we sent an email invitation (as well as additional reminder emails) to another set of 20 policymakers to attend an online policy briefing where we gave a presentation on gender and policymaking. At the end of the briefing, we sent them the link to the pledge website. Participants were randomly assigned to one of two groups. In the control group, the email contained an invitation to attend an online policy briefing and a Zoom link. In the treatment group, the email also contained two personal stories of how women have been impacted by policies. Out of 10 participants in the treatment group, one participant attended the briefing, but none signed the pledge. No participants from the control group attended the briefing or signed the pledge. While response rate and attendance were low, we were encouraged that at least one policymaker attended the briefing. Thus, we chose to focus on attendance rate as our primary behavioral measure based on these pilots.

## Methods

### Participants

We first did a power analysis to determine our sample size. Assuming a minimum effect size w = 0.25, alpha = 0.05, power = 0.95, we need a minimum number of 208 participants in total. Thus, we recruited a group of 276 policymakers and policy influencers from the Northwest Territories in Canada. We obtained participant emails from public records, and prior direct contact with them unrelated to this study. Of the 276 participants, 208 were elected officials (i.e., Chiefs, City Councillors, Mayors, Members of the Legislative Assembly) and 68 were policy influencers who work closely with policymakers (i.e., Deputy Ministers, Assistant Deputy Ministers, cabinet policy advisors, senior policy advisors, and policy analysts). The policy influencers were important to include in the study because they often directly influence the policy decisions of elected officials with whom they closely work.

Participants were required to meet the following criteria to be included in our study: they must be involved in developing, drafting, or influencing policy and/or legislation within the NWT; they must have a publicly available email address; and they must speak English and is at or over the age of 18. Participants were excluded from our study based on the following criteria: if they actively opted-out or withdrew from the study; if they had an invalid email address; if we received an email confirmation that the participant had retired from their position; and if they participated in a group opposite to that assigned by random assignment. In total eight participants from the control group and five from the treatment group were excluded from our study due to invalid email addresses and/or job retirement, leaving a final sample size of 263. There were 133 participants in the treatment group (100 elected officials, 33 policy influencers; 43% Female) and 130 participants in the control group (97 elected officials, 33 policy

 

influencers; 44% Female). Participants were informed about the study in the email invitations (outlined below) and consent was implied by default through the reception of these emails. Participants were able to opt-out of the study by responding to the email to withdraw their consent. The study was approved by the UBC Behavioural Research Ethics Board, with all methods performed in accordance with relevant guidelines and regulations.

### Stimuli and procedure

Participants were randomly assigned to the treatment condition or the control condition. Participants in both conditions received a personalized email invitation from the Status of Women Council of the NWT inviting them to attend a 10-minute online briefing session on gender equity and policy. The email invitation for the treatment condition contained two personal stories highlighting the disproportionate impact of policies for two women. The email invitation for the control condition did not contain personal stories. To maximize attendance, all invitations contained a link to a Doodle poll, where participants could indicate their choice of a briefing session from a list of five alternate dates and times if they couldn't make the briefing on the default date. See SI briefing email invitation in the control and treatment conditions.

The study ran for a total of three weeks from April 21 to May 13, 2022. Email invitations were sent to both groups on April 21, 2022 with three email reminders sent over the subsequent two weeks for both groups. The briefing session was a 10-minute presentation on gender equity in policymaking run by the Status of Women Council of the NWT. The briefing was identical in both conditions except that the briefing in the treatment condition contained personal stories (as in the email invitation). See SI for the briefing slides. The briefing was prerecorded to ensure that all participants in a given condition were given the same information. All questions and comments from the participants were recorded and responded to with a scripted response indicating that they would be contacted with a personal follow-up after the briefing to answer their questions. Attendance reports were generated over Zoom and cross-checked against the participant list, to keep track of who attended the briefing in each condition.

At the end of each briefing, participants in both conditions were sent a link to a website (https://www.noeconomicabuse.com/) where they could sign a public pledge to lead and advocate for equity-oriented policymaking. The website also contained information and tools that policymakers could use to implement the pledge in their policy work. At the end of the study, all participants were sent an email on May 13, 2022 thanking them for their participation with a link to the website to sign the pledge.

## Results

### Pre-registered analysis

To address our pre-registered hypothesis (https://osf.io/ht3yn), we conducted a chi-square test and found a marginally significant difference in attendance rate between the two conditions [$X^2(1,263) = 2.86$, $p = .09$]. In the treatment condition, 3.0% of participants (4 out of 133) attended the briefing, whereas 7.7% of participants (10 out of 130) in the control condition attended the briefing (Fig 1). The result was opposite to our hypothesis, suggesting a small backfiring effect.

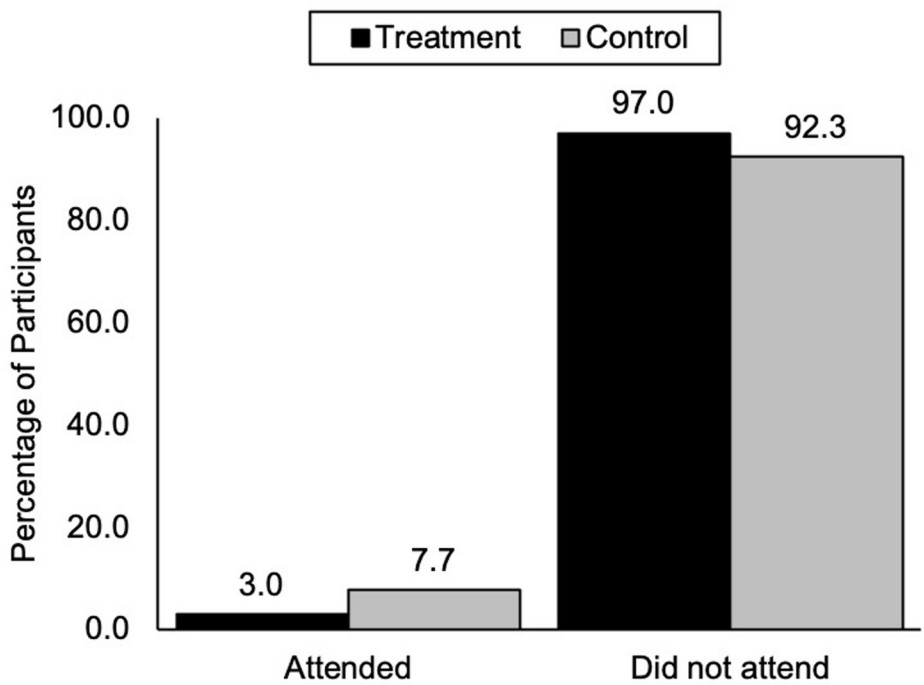

**Fig 1. Briefing attendance in the control and treatment conditions.**

### Exploratory analyses

To understand this backfiring effect, we conducted exploratory analyses to test whether the number of participants who accepted the invitation to attend the briefing, and actually did attend, was different between treatment and control conditions. A chi-square test showed a significant difference between the two conditions [$X^2(1,28) = 5.14$, $p = .02$]. A total of 14 participants in the treatment condition accepted the invitation, 4 attended the briefing (28.6%) and 10 did not (71.4%). A total of 14 participants in the control condition accepted the invitation, 10 attended the briefing (71.4%) and 4 did not (28.6%, Fig 2). Among the 14 participants in the treatment condition who accepted the invitation, 9 were elected officials, while 5 were policy influencers. Among the 14 participants in the control condition who accepted the invitation, 7 were elected officials, while 7 were policy influencers.

We also examined the response rate (accepted or declined) to the email invitation but found no significant difference between the two conditions [$X^2(1,263) = 0.13$, $p = .72$]. In the treatment condition, 21.8% of participants (14 accepted, 15 declined) responded to the invitation whereas 20.0% of participants in the control condition (14 accepted, 12 declined) responded (Fig 3).

We ran a final chi-square test to see whether there was a difference in pledge signing. In the treatment condition, 1.5% of participants (2 out of 133) signed the pledge (Fig 4). This included a Chief and a Yellowknife City Councillor, only one of which accepted the briefing invitation but neither attended the briefing. In the control condition, 0.8% of participants (1 out of 130) signed the pledge. This participant was a government equity officer who accepted the invitation and attended the briefing. However, the signing rate was not statistically significant between the two conditions [$X^2(1,263) = 0.31$, $p = .57$].

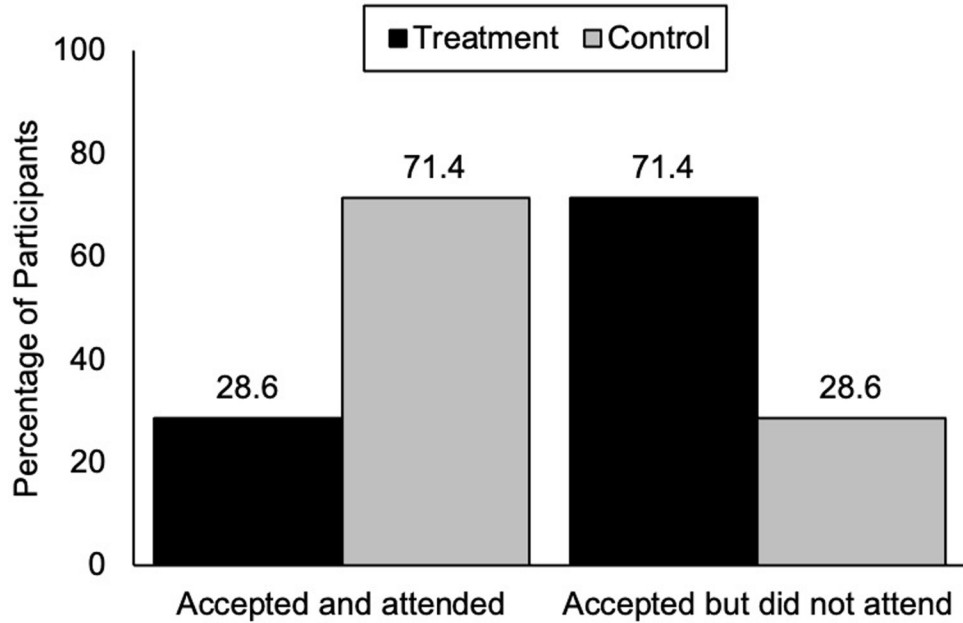

**Fig 2. Invitation acceptance and briefing attendance in control and treatment conditions.**

## Qualitative follow-up analysis

Following the completion of our study, the Status of Women Council of the NWT conducted follow-up phone calls with participants who did not respond to the email invitation. Out of 208 participants who did not respond to the invitation, the Council was able to make direct contact with 47 participants. These participants were asked why they did not respond to the

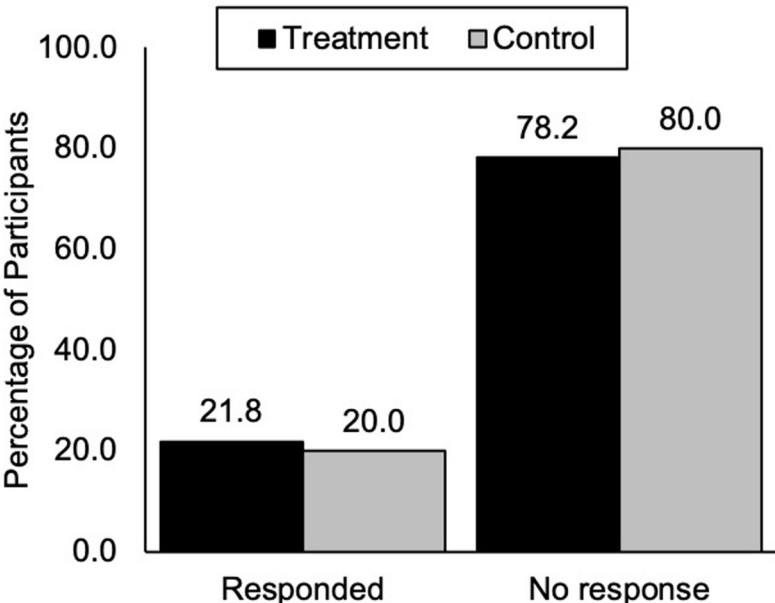

**Fig 3. Email response in the control and treatment conditions.** Email response includes both accepted and declined invitations.

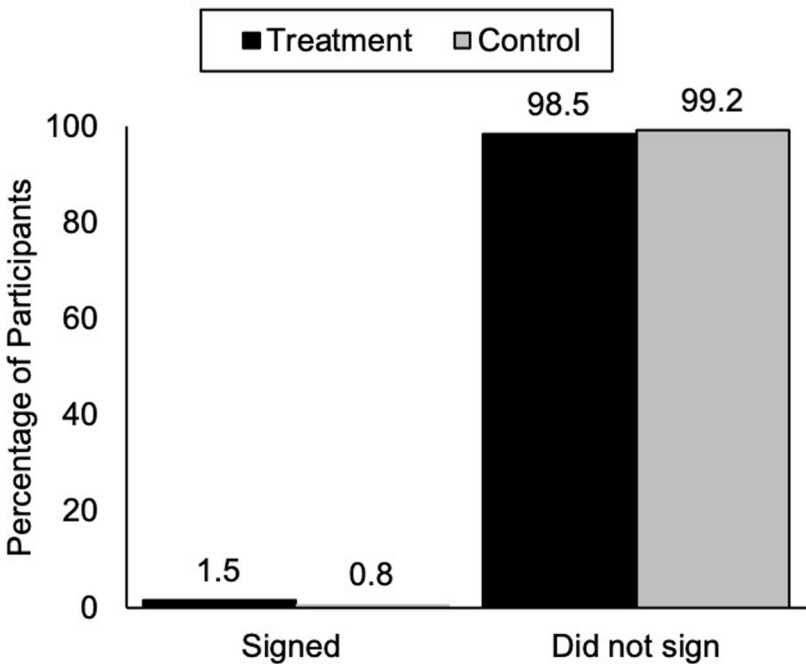

**Fig 4. Pledge signing in the control and treatment conditions.**

email invitation, resulting in qualitative data that was examined for common themes and used to understand why participants did not respond. In descending order of frequency, participants responded that they were: (1) not available on the briefing date/time and didn't know there were alternative dates/times; (2) out of office on travel for their job; (3) couldn't attend due to last-minute changes in priorities; (4) not available because the briefings were scheduled during the NWT land hunting season; and (5) their emails were monitored by an assistant who didn't have the authority to accept or decline on the participant's behalf (Yellowknife City Councillors only).

In addition to the follow-up phone calls, several participants reached out to the Council throughout the course of the study to provide anecdotal, positive responses. One community councillor who attended the briefing felt that these issues were very important and should be brought up at a community level. One participant noted that they were "really glad you guys are offering this, and [I] have recommended it to my teammates." Another participant who declined the email invitation but said they had "taken GBA+ (Gender-based Analysis Plus) and it changed how I consider policymaking. I am glad you are doing this." Another participant, a Member of the Legislative Assembly (MLA) who didn't respond to the invitation, directly reached out to the Council to say that anytime the Council wants to bring anything related to the legislature, the MLA would be willing to sponsor it. The Council also received several direct requests for additional information, training and support for gender equity assistance because of this study.

## General discussion

The goal of the current study was to test the effectiveness of personal stories to nudge policymakers from the NWT to attend a policy briefing on gendered impacts of policy. Overall, we found that using personal stories in the email invitation was not successful in increasing attendance at the briefings, response rate to the invitations, or pledge signing. In fact, our results

were the opposite to our hypothesis regarding attendance rate, suggesting a backfiring effect in which the control group was more likely to attend the briefing than the treatment group. These results contrast with the previously reviewed literature, which generally suggest a persuasive benefit to personal stories. However, few of these studies contained pre-registered hypotheses as ours did, potentially alerting to the presence of bias in this literature. The current results suggest the need for caution in the use of personal stories and reveal their limitations to change behaviors among policymakers.

There were several possible reasons for the backfiring effect. First, we found that the treatment and control conditions differed in the number of elected officials (vs. policy influencers) who accepted the invitation to the briefing. This may be due to the fact that several communities in the NWT experienced severe flooding during our study and many elected officials in these communities had to prioritize community evacuations over the briefing. These elected officials just happened to be in the treatment condition. The imbalance of policymakers between conditions despite random assignment could explain the finding that fewer participants in the treatment condition attended the briefing. Second, two communities in the NWT held local elections and some elected officials in the treatment condition were away on duty travel during our study. This likely limited the available time that elected officials would have to attend our briefing. Given that the treatment group had a greater proportion of elected officials who accepted the invitation, but ultimately did not attend, these two external factors may explain the backfiring effect in attendance rate. This is supported by qualitative data collected from our follow-up phone calls with participants, which showed that travel and changing priorities were the second and third most frequently stated reasons for not attending the briefing.

The top reason was that participants did not know that alternative dates/times were available, suggesting that the poll—which gave them the option to choose from five alternate dates/times—was not sufficiently salient to capture attention. The presence of alternatives may have been particularly obscured in the treatment group emails, in which the addition of personal narratives placed the note about alternative dates/times much further into the email than in the control group. Given that effective communication with policymakers likely depends upon succinct language, this increased length may explain why the treatment produced the opposite of the hypothesized effect. Finally, policymakers may find personalized stories detract from the larger structural cause of the problem, and therefore are more likely to ignore the briefing. Future studies could use depersonalized stories presented from the perspective of constituent groups, rather than individuals, and see whether collective stories are more effective in motivating action from policymakers.

Designing an effective intervention that influences policymakers on gender equity issues would likely take some time. Qualitative data collected after the study reveals some insights on how to design such an intervention, such as increasing the salience of alternative briefing sessions. Overall, we received a number of anecdotal comments from participants regarding the importance of the issue and offering additional support, suggesting that the study was well received among participants. This indicates that many policymakers may already be aware of the importance of the issue and may be looking for ways to improve gender equity in the NWT. Future research can consider providing policymakers with more tangible ways that gender equity can be incorporated into their policy work, in addition to a policy briefing or a public pledge.

Additionally, we note that the participants in the treatment condition who signed the public pledge did so without attending the briefing, signing only after receiving a thank-you email with a link to the pledge website. Considering the many studies that found pledges to be effective at promoting target behaviors [15–20], encouraging policymakers to make a public commitment toward prioritizing gender equity in policy is likely a worthwhile endeavour. Future

research should explore the possibility of using depersonalized stories to nudge policymakers to commit to gender equity work in the absence of a policy briefing.

Finally, due to the small, interconnected nature of the political community in the NWT, the potential for contamination between conditions as a result of sharing emails was a possibility. We attempted to mitigate this possibility by including a statement in the email invitation discouraging forwarding the email invite to other people. While we did not encounter contamination between conditions in the study, we did notice an interesting phenomenon in which several uninvited guests, who were not on the original participant list, responded to the email invitation, or actually attended a briefing. While these uninvited guests were not included in our data analysis, this may reveal a 'chain of influence', allowing us to track how information and issues are disseminated throughout the NWT political community, and could be an interesting avenue for future research.

While the current study yielded null results, it is important to publish these findings, especially backfiring results, for two reasons. First, the null results provide an empirical contribution to show that this particular intervention did not produce the predicted effect in this particular population, such that future studies can improve the study design. Second, the study provides a theoretical contribution to show the limits of this intervention, revealing the unique barriers faced by this particular population of policymakers that are different from past study populations. As the qualitative data suggest, policymakers may be more impacted by external factors (e.g., flooding, local elections) then personal stories.

## Supporting information

**S1 File. Briefing email invitations in the control and treatment conditions, and briefing slides.**
(DOCX)

## Author Contributions

**Conceptualization:** Lindsay Blair Bochon, Janet Dean, Tanja Rosteck, Jiaying Zhao.

**Data curation:** Lindsay Blair Bochon, Janet Dean, Tanja Rosteck, Jiaying Zhao.

**Formal analysis:** Lindsay Blair Bochon, Janet Dean, Tanja Rosteck, Jiaying Zhao.

**Investigation:** Lindsay Blair Bochon, Janet Dean, Tanja Rosteck, Jiaying Zhao.

**Methodology:** Lindsay Blair Bochon, Janet Dean, Tanja Rosteck, Jiaying Zhao.

**Project administration:** Lindsay Blair Bochon, Janet Dean, Tanja Rosteck, Jiaying Zhao.

**Supervision:** Jiaying Zhao.

**Visualization:** Lindsay Blair Bochon.

**Writing – original draft:** Lindsay Blair Bochon.

**Writing – review & editing:** Lindsay Blair Bochon, Janet Dean, Tanja Rosteck, Jiaying Zhao.

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
