## [Decision Letter · Decision Letter 0]

14 Jul 2023

PONE-D-23-14469Nudging policymakers on the gendered impacts of policyPLOS ONE

Dear Dr. Zhao, Thank you for submitting your manuscript to PLOS ONE. After careful consideration, we feel that it has merit but does not fully meet PLOS ONE’s publication criteria as it currently stands. Therefore, we invite you to submit a revised version of the manuscript that addresses the points raised during the review process.

We look forward to receiving your revised manuscript.

Kind regards,

Hidenori Komatsu

Academic Editor

PLOS ONE

Journal Requirements:

2. Please amend either the title on the online submission form (via Edit Submission) or the title in the manuscript so that they are identical.

**Additional Editor Comments:**

The following two points are especially important for revision:Please consider carefully how you define your interventions (nudging vs. priming) according to Reviewer #1's comments.Also please describe details of the qualitative analysis more thoroughly (i.e., methodology, data,  relationship with the statistical analysis etc.) according to Reviewer #2's comments.

Reviewers' comments:

Reviewer's Responses to Questions

**Comments to the Author**

1. Is the manuscript technically sound, and do the data support the conclusions?

Reviewer #1: Yes

Reviewer #2: Yes

2. Has the statistical analysis been performed appropriately and rigorously? 

Reviewer #1: I Don't Know

Reviewer #2: Yes

3. Have the authors made all data underlying the findings in their manuscript fully available?

Reviewer #1: Yes

Reviewer #2: Yes

4. Is the manuscript presented in an intelligible fashion and written in standard English?

Reviewer #1: Yes

Reviewer #2: Yes

5. Review Comments to the Author

Reviewer #1: Thank you for the opportunity to review this manuscript. I have some concerns but overall I conclude this study is likely professionally executed and provides a well-evidenced challenge to some of the enthusiasm around what I characterise as ‘priming’ interventions to achieve social goals.

My suggestions are to ensure that the result of the study fit with the wider literature and contribute to the broader debate.

First, I think this project would be better framed as a test of ‘priming policymakers’ rather than nudging policymakers. Why? Nudging is classically a libertarian paternalist notion. The key features of a nudge as introduced are:

1. The agency setting out the manipulation/intervention has some sort of authoritative position – as a governing body with a jurisdiction over the target or as an employer with some presumed interest the wellbeing of the target

2. The aim of the intervention is to improve the welfare of the target (as they would realistically or ideally see it themselves)

3. The targeted agent has the choice to do other than what the manipulation tries to get them to do (i.e. its not ‘hard paternalism’ or some sort of legal or contractual requirement)

See: Epstein RA (2018) The Dangerous Allure of Libertarian Paternalism. Review of Behavioral Economics 5(3–4): 389–416. DOI: 10.1561/105.00000087.

While 3. applies here (plenty of policymakers ignored the email and did not sign the pledge ultimately), points 1. and 2. do not apply. This is effectively a strategic communication or act of lobbying between actors that do not hold any specific authority over one another. Policymakers are constantly subject to multiple attempts to manipulate their behavior and priorities. The intervention is intended to contribute to the public good, not the private welfare of the policymaker. It is not a nudge: it is a prime.

The advantage of this framing is 1) it is more precise in terms of terminology; and 2) it highlights how much broader this result is as it challenges not only nudges but a whole category of manipulations of which some nudges are merely a sub-category.

Second, this point could be further strengthened by more attention to the literature that led to the hypothesis generation for this experiment. How many of the studies discussed in the previous literature, I am thinking particularly in the Braddock & Dillard meta-analysis, involved pre-registered studies? If pre-registration has been uncommon in this area, then this might indicate various forms of publication or researcher bias have produced the previous results and this study (arguably superior in research design) is alerting behavioural scientists to be cautious. Pre-registration is a strength of this study that is worth emphasising against the general quality of the literature.

Smaller related points:

‘It is time to reorient the focus of nudge onto government itself — nudging policymakers directly to improve the way that policy is made’

Besides the use of nudge which I think is better re-characterised, this seems to involve unnecessary editorialising in a scientific study. The results of the study suggest that now might not be the time reorient narrative priming onto policymakers because here is some evidence it does not do all that much good. So reframing this as a question would show more alignment with the project.

*

Could the authors be more explicit when and how the policymakers consented to participate in the study? It would be good to know the full context about their knowledge they had about what was going on during the study.

*

It could be worth caveating that one of the possible reasons for the treatment producing the opposite to the theorised effect is that it places the note about alternative times to see the webinar a lot further into the email than in the control group. I imagine the effectiveness of communications with policymakers depends a great deal on brevity and user-friendliness. The narratives might have disrupted these things. I do not see this as a direct problem for the results of the study as, if true, it merely indicates that narrative persuasion could be useful but not compared to more fundamental aspects of effective communication. Hence, it is another way of explaining the negative/null results.

Reviewer #2: Overall reflections

The paper provides interesting results of a policy evaluation process that is not often reported in policy papers. I commend the authors for reporting negative results. Below I detail what could be done to strengthen the paper.

Specific comments

Comment 1: In line 58 in the introduction section page 3. The authors appear to suggest that gender equity has been overlooked by policymakers due to high information-processing demands. I believe this is not the only issue that may propagate inequities in gender. I propose that they include other reasons why gender equity is overlooked in addition to the issue of information.

Comment 2: In terms of the pilot study, I wondered why the authors did not conduct a small qualitative study to explore the drivers behind the results depicted. The authors claim that failure to sign the pledge was because of a lack of sufficient information without any evidence. It would have been beneficial to have some qualitative component to explore the reasons and use that to design the second pilot.

Comment 3: Based on the second pilot, I am wondering whether the choice of attendance rate as the primary behavioral measure should have been an intermediate outcome and retain the signing as the end result.

Comment 4: The authors reported that the responses were scripted, could the authors explain further, what this means? What if the participants asked a question that was outside the scripted responses? How was it handled?

Comment 5: were the participants who attended the policy briefing given any support on how to use the tool to implement the pledge in their policy work? This could be a gap that requires some attention.

Comment 6: One methodological issue I have with the authors is the scanty details provided on the qualitative component. I think they should detail the analysis process, the selection process of those whom they interviewed. There was also very limited information on the results with a tendency to present the summary quantitatively.

Comment 7: In the discussion section, page 13 the authors report that the control group was more likely to attend the briefing than the treatment group. Although they provide information that was linked to context such as flooding etc, can the authors give an indication of the distribution of the 47 participants interviewed qualitatively from either control and intervention groups? This will provide a better sense of the reasons provided.

Comment 8: In page 14, the authors report on another qualitative conducted after the study, was this a different one from what was reported in this paper? If yes I suggest a better integration of the qualitative data throughout the paper with a detailed process of analysis

6. PLOS authors have the option to publish the peer review history of their article (what does this mean?). If published, this will include your full peer review and any attached files.

Reviewer #1: **Yes: **Nick Cowen

Reviewer #2: No

---

## [Author Response · Author response to Decision Letter 0]

9 Aug 2023

Please see the attached Response to Reviewers letter.

---

## [Decision Letter · Decision Letter 1]

21 Sep 2023

PONE-D-23-14469R1Nudging policymakers on gendered impacts of policyPLOS ONE

Dear Dr. Zhao,

Thank you for submitting your manuscript to PLOS ONE. After careful consideration, we feel that it has merit but does not fully meet PLOS ONE’s publication criteria as it currently stands. Therefore, we invite you to submit a revised version of the manuscript that addresses the points raised during the review process.

We look forward to receiving your revised manuscript.

Kind regards,

Hidenori Komatsu

Academic Editor

PLOS ONE

Journal Requirements:

Additional Editor Comments:

There are 2 comments from me as an Academic Editor.

First, please make sure that the consistency of the floating points through the manuscript including all the descriptions and figures. More specifically, the order of magnitude is 0.1 in Fig 4 while integers are used in Fig 1, 2, and 3. The order of magnitude in Fig 1, 2, and 3 should be 0.1 too.

Second, this is more important. I checked your data uploaded in OSF (https://osf.io/85fg6 last modified, May 11, 2023) by myself and might find some error in your analysis. Given that 'Condition_1_control_2_treatment_' means the treatment group if the value is 2, responses are excluded if 'Response' is 'Invalid', and the respondents signed the pledge if 'SignedPledge' is 'Y', the number of participants who signed the pledge is 0 according to my analysis, although your result is 2.

'Line 269: In the treatment condition, 1.5% of participants (2 out of 133) signed the pledge (Fig 4). This included a Chief and a Yellowknife City Councillor, only one of which accepted the briefing invitation but neither attended the briefing.'

Also, if there were 3 respondents who signed the pledge in total (2 in the treatment group and 1 in the control group), we could see 3 cells filled with 'Y' or similar, but I can see only 1 cell of 'Y' in the 4th row (the participant ID is 3853). I might misunderstand something but please clarify why this happens and correct your manuscript if this is an error.

For your information, I just copy-paste the results of my analysis of your data using Matlab:

Contents

Fig 1

Fig 2

Fig 3

Fig 4

data=readtable('NudgingPolicymakersData.xlsx');

Warning: Column headers from the file were modified to make them valid MATLAB

identifiers before creating variable names for the table. The original column

headers are saved in the VariableDescriptions property.

Set 'VariableNamingRule' to 'preserve' to use the original column headers as

table variable names.

Fig 1

height(data(data.Condition_1_control_2_treatment_==1 & (strcmp(data.Response,'Invalid') == false),:)) % valid control 130, confirmed

height(data(data.Condition_1_control_2_treatment_==1 & (strcmp(data.Response,'Invalid') == false) & (strcmp(data.Attendance_Y_N_,'Y') | strcmp(data.Attendance_Y_N_,'y')),:)) % valid control attended 10, confirmed

height(data(data.Condition_1_control_2_treatment_==2 & (strcmp(data.Response,'Invalid') == false),:)) % valid treatment 133, confirmed

height(data(data.Condition_1_control_2_treatment_==2 & (strcmp(data.Response,'Invalid') == false) & (strcmp(data.Attendance_Y_N_,'Y') | strcmp(data.Attendance_Y_N_,'y')),:)) % valid treatment attended 4, confirmed

ans =

130

ans =

10

ans =

133

ans =

4

Fig 2

height(data(data.Condition_1_control_2_treatment_==1 & (strcmp(data.Response,'Invalid') == false) & (strcmp(data.Response, 'Accepted')) ,:)) % valid control accepted 14, confirmed

height(data(data.Condition_1_control_2_treatment_==1 & (strcmp(data.Response,'Invalid') == false) & (strcmp(data.Attendance_Y_N_,'Y') | strcmp(data.Attendance_Y_N_,'y')),:)) % valid control attended 10, confirmed

height(data(data.Condition_1_control_2_treatment_==2 & (strcmp(data.Response,'Invalid') == false) & (strcmp(data.Response, 'Accepted')) ,:)) % valid treatment accepted 14, confirmed

height(data(data.Condition_1_control_2_treatment_==2 & (strcmp(data.Response,'Invalid') == false) & (strcmp(data.Attendance_Y_N_,'Y') | strcmp(data.Attendance_Y_N_,'y')),:)) % valid treatment accepted and attended 4, confirmed

ans =

14

ans =

10

ans =

14

ans =

4

Fig 3

height(data(data.Condition_1_control_2_treatment_==1 & (strcmp(data.Response,'Invalid') == false) & (strcmp(data.Response, 'Accepted')) ,:)) % valid control accepted 14, confirmed

height(data(data.Condition_1_control_2_treatment_==1 & (strcmp(data.Response,'Invalid') == false) & ((strcmp(data.Response, 'Decline') | (strcmp(data.Response, 'Declined')))) ,:)) % valid control declined 12, confirmed

height(data(data.Condition_1_control_2_treatment_==2 & (strcmp(data.Response,'Invalid') == false) & (strcmp(data.Response, 'Accepted')) ,:)) % valid treatment accepted 14, confirmed

height(data(data.Condition_1_control_2_treatment_==2 & (strcmp(data.Response,'Invalid') == false) & ((strcmp(data.Response, 'Decline') | (strcmp(data.Response, 'Declined')))) ,:)) % valid treatment declined 15, confirmed

ans =

14

ans =

12

ans =

14

ans =

15

Fig 4

height(data(data.Condition_1_control_2_treatment_==1 & (strcmp(data.Response,'Invalid') == false),:)) % valid control 130, confirmed

height(data(data.Condition_1_control_2_treatment_==1 & (strcmp(data.Response,'Invalid') == false) & (strcmp(data.SignedPledge,'Y') | strcmp(data.SignedPledge,'y')),:)) % valid control attended, confirmed

height(data(data.Condition_1_control_2_treatment_==2 & (strcmp(data.Response,'Invalid') == false),:)) % valid treatment 133, confirmed

height(data(data.Condition_1_control_2_treatment_==2 & (strcmp(data.Response,'Invalid') == false) & (strcmp(data.SignedPledge,'Y') | strcmp(data.SignedPledge,'y')),:)) % valid treatment attended, actually 0 but should be 2?

unique(data.SignedPledge) % there are only 'Y' and blank cells.

sum(strcmp(data.SignedPledge,'Y')) % actually 1 but should be 3?

ans =

130

ans =

1

ans =

133

ans =

0

ans =

2×1 cell array

{0×0 char}

{'Y' }

ans =

1

Reviewers' comments:

Reviewer's Responses to Questions

**Comments to the Author**

1. If the authors have adequately addressed your comments raised in a previous round of review and you feel that this manuscript is now acceptable for publication, you may indicate that here to bypass the “Comments to the Author” section, enter your conflict of interest statement in the “Confidential to Editor” section, and submit your "Accept" recommendation.

Reviewer #1: All comments have been addressed

Reviewer #2: All comments have been addressed

2. Is the manuscript technically sound, and do the data support the conclusions?

Reviewer #1: Yes

Reviewer #2: Yes

3. Has the statistical analysis been performed appropriately and rigorously? 

Reviewer #1: I Don't Know

Reviewer #2: Yes

4. Have the authors made all data underlying the findings in their manuscript fully available?

Reviewer #1: Yes

Reviewer #2: Yes

5. Is the manuscript presented in an intelligible fashion and written in standard English?

Reviewer #1: Yes

Reviewer #2: Yes

6. Review Comments to the Author

Reviewer #1: (No Response)

Reviewer #2: I think the authors have addressed all the issues i raised. However, i feel like the issueof analysis of qualitative data is not fully convincing in terms of access to raw data.

The analysis of qualitative data is often not well-presented making critics not value the rigorous process that qualitative is managed. Other than that, i would have preferred a better description of how that data was managed and themes that emerged. This is a lesson that needs to be emphasized to reinforce the rigor that is needed to maintain the role qualitative data play in generating evidence.

7. PLOS authors have the option to publish the peer review history of their article (what does this mean?). If published, this will include your full peer review and any attached files.

Reviewer #1: **Yes: **Nick Cowen

Reviewer #2: **Yes: **Timothy Abuya

---

## [Editor Report · Decision Letter 2]

4 Oct 2023

Nudging policymakers on gendered impacts of policy

PONE-D-23-14469R2

Dear Dr. Zhao,

We’re pleased to inform you that your manuscript has been judged scientifically suitable for publication and will be formally accepted for publication once it meets all outstanding technical requirements.

Kind regards,

Hidenori Komatsu

Academic Editor

PLOS ONE

---

## [Editor Report · Acceptance letter]

10 Oct 2023

PONE-D-23-14469R2 

Nudging policymakers on gendered impacts of policy 

Dear Dr. Zhao:

I'm pleased to inform you that your manuscript has been deemed suitable for publication in PLOS ONE. Congratulations! Your manuscript is now with our production department. 

Kind regards, 

on behalf of

Dr. Hidenori Komatsu 

Academic Editor

PLOS ONE